# Sol–Gel Synthesis and Characterization of Coatings of Mg-Al Layered Double Hydroxides

**DOI:** 10.3390/ma12223738

**Published:** 2019-11-13

**Authors:** A. Smalenskaite, M. M. Kaba, I. Grigoraviciute-Puroniene, L. Mikoliunaite, A. Zarkov, R. Ramanauskas, I. A. Morkan, A. Kareiva

**Affiliations:** 1Department of Inorganic Chemistry, Faculty of Chemistry, Vilnius University, Vilnius LT-03225, Lithuania; aurelija.smalenskaite@gmail.com (A.S.); inga.grigoraviciute@gmail.com (I.G.-P.); lina.mikoliunaite@chf.vu.lt (L.M.); aleksej.zarkov@chf.vu.lt (A.Z.); 2Department of Chemistry, Institute of Natural Sciences, Bolu Abant Izzet Baysal University, 14030 Bolu, Turkey; musakaba136@yahoo.com (M.M.K.); morkan_i@ibu.edu.tr (I.A.M.); 3Center for Physical Sciences and Technology, LT-10257 Vilnius, Lithuania; rimantas.ramanauskas@ftmc.lt

**Keywords:** layered double hydroxides, Mg-Al, sol–gel synthesis, coatings, spin coating, silicon, stainless steel

## Abstract

In this study, new synthetic approaches for the preparation of thin films of Mg-Al layered double hydroxides (LDHs) have been developed. The LDHs were fabricated by reconstruction of mixed-metal oxides (MMOs) in deionized water. The MMOs were obtained by calcination of the precursor gels. Thin films of sol–gel-derived Mg-Al LDHs were deposited on silicon and stainless-steel substrates using the dip-coating technique by a single dipping process, and the deposited film was dried before the new layer was added. Each layer in the preparation of the Mg-Al LDH multilayers was separately annealed at 70 °C or 300 °C in air. Fabricated Mg-Al LDH coatings were characterized by X-ray diffraction (XRD) analysis, scanning electron microscopy (SEM), and atomic force microscopy (AFM). It was discovered that the diffraction lines of Mg_3_Al LDH thin films are sharper and more intensive in the sample obtained on the silicon substrate, confirming a higher crystallinity of synthesized Mg_3_Al LDH. However, in both cases the single-phase crystalline Mg-Al LDHs have formed. To enhance the sol–gel processing, the viscosity of the precursor gel was increased by adding polyvinyl alcohol (PVA) solution. The LDH coatings could be used to protect different substrates from corrosion, as catalyst supports, and as drug-delivery systems in medicine.

## 1. Introduction

Layered double hydroxides (LDHs) are compounds composed of positively charged brucite-like layers, with an interlayer gallery containing charge-compensating anions and water molecules. The metal cations occupy the centres of shared oxygen octahedra whose vertices contain hydroxide ions that connect to form infinite two-dimensional sheets [1,2,3,4,5,6]. A chemical formula of Mg-Al LDH can be expressed as [Mg^2+^_1-x_Al^3+^_x_(OH)_2_]^x+^(A^m-^)_x/m_]·nH_2_O, where A^m−^ is an intercalated anion.

LDHs are widely used in commercial products as adsorbents, catalysts, flame retardants, osmosis membranes, energy-storage materials, and sensors [3,7,8,9,10,11,12,13]. LDH materials have been successfully used for drug and gene delivery, cosmetics, cancer therapy, biosensing, and as antibacterial agents [14,15,16,17,18,19]. LDHs have been studied for their potential application to the removal of anions and also toxic metal ions from contaminated waters [20,21,22,23,24,25,26]. In recent years, inorganic–organic hybrid luminescence materials have been widely investigated due to their novel properties of forming stable compounds with lanthanides in the interlayer space of LDHs [4,6,27,28,29]. The LDH layers were demonstrated to offer anticorrosion protection [30,31,32,33,34].

There are many general methods for the preparation of bulk LDHs, such as co-precipitation [2,35,36], sol–gel synthesis [4,5,37,38], urea hydrolysis [39,40], hydrothermal synthesis [41], and others [38,42,43]. Several synthesis methods were suggested for the fabrication of LDH coatings on different substrates. In [44,45,46,47], facile in situ growth and dispersing methods were used to prepare anticorrosive LDH films on the surfaces of different Al and Mg alloys. The LDH-sealing layers and coatings on anodic aluminium oxide, titanium dioxide, aluminium, steel alloys, and other metal substrates were also prepared using aqueous solution, hydrothermal, co-precipitation, or hybrid hydrothermal–co-precipitation methods [48,49,50,51,52,53,54,55]. Wu et al. [56] suggested the use of a urea hydrolysis method for the synthesis of LDH films on Al alloy. The urea-assisted synthetic approach was transferred for the fabrication of LDH coatings on a plasma electrolysis (PE) Al alloy coating [57]. Recently, formation processes for LDH coatings on Mg alloy or on alumina by the CO_2_ pressurization and electrophoretic deposition methods, respectively, have been developed [58,59].

It is well-known that the sol–gel processing route for the preparation of thin films of different materials is a low-cost and simple method, which allows for better chemical homogeneity due to molecular-level mixing of the precursors [60,61,62,63,64,65]. However, the fabrication of LDH films by the sol–gel chemistry method has not been given sufficient attention to date. The number of such studies is rather limited, with only one publication [66]. Moreover, the authors of this study provided only the results about the preparation of amorphous Mg–Al–Eu–O thin films on silica glass substrates by a sol–gel dip-coating method with a heat treatment at 700 °C. Thus, no evidence of the formation of LDHs during sol–gel processing was documented and characterized. Therefore, the present study will discuss for the first time the stabilization of LDH films grown using a sol–gel synthetic approach on silicon and stainless-steel substrates by the dip-coating technique. The LDHs were fabricated by reconstruction of mixed-metal oxides (MMOs) in deionized water. The MMOs were obtained by calcination of the precursor gels.

## 2. Experimental

The Mg_3_Al LDH specimens were prepared by the sol–gel technique using metal nitrates Mg(NO_3_)_2_·6H_2_O (99.9%, Fluka, Saint Louis, MO, USA) and Al(NO_3_)_3_·9H_2_O (99.9%, Fluka, Saint Louis, MO, USA) dissolved in 50 mL of deionized water as starting materials. To the obtained mixture, a 0.2 M solution of citric acid (C_6_H_8_O_7_, 99.0%, Alfa Aesar, Haverhill, MA, USA) was added. The resulting solution was additionally stirred for 1 h at 80 °C. Finally, 2 mL of ethylene glycol (C_2_H_6_O_2_, 99.0%, Alfa Aesar, Haverhill, MA, USA) was added with continued stirring at 150 °C. During the evaporation of solvent, the transformations from the sols to the gels occurred. The synthesized precursor gels were dried at 105 °C for 24 h. The MMOs were obtained by heating the gels at 650 °C for 4 h. LDHs were fabricated by reconstruction of MMOs in deionized water at 80 °C for 6 h. The relative humidity of the atmosphere was about 50%. Mg_3_Al LDH coatings were synthesized using the sol–gel method in different solutions. In the first attempt, only an aqueous solution of LDH was used. Secondly, 0.5 g of LDH was mixed with 1 g of polyvinyl alcohol (PVA) (PVA7200, 99.5%, Aldrich, Saint Louis, MO, USA) in distilled water. LDH suspensions were deposited on silicon and stainless-steel substrates using the dip-coating technique by a single dipping process, and the deposited film was dried before the new layer was added.

X-ray diffraction (XRD) analysis of synthesized compounds was performed with a MiniFlex II diffractometer (Rigaku, The Woodlands, TX, USA) using primary-beam Cu Kα radiation (λ = 1.541838 Å). The 2θ° angle of the diffractometer was tuned from 8 to 80° in steps of 0.02°, with the measuring time of 0.4 s per step. The surface morphological features were characterized using a scanning electron microscope (SEM) (Hitachi SU-70, Tokyo, Japan). The roughness of the Mg_3_Al LDH films was estimated using an atomic force microscope (AFM) (BioscopeII/Catalyst, Karlsruhe, Germany). The ScanAsyst, operated in the peak-force tapping mode and equipped with a wafer of silicon nitride probe Asyst at the air AFM tip, was used for imaging. Surface root mean square (RMS) values were calculated using the MATLAB R2015b programme.

## 3. Results and Discussion

The XRD patterns of Mg_3_Al LDHs obtained from the silicon and steel substrates are presented in Figure 1 and Figure 2, respectively. Evidently, the intensity of Si reflection originating from the substrate is much higher in comparison to the main reflection of LDH samples. However, after eliminating silicon reflection from the XRD patterns (see the insertion in Figure 1), the main reflections clearly represent the formation of an LDH structure after 15 dipping procedures. The formation of Mg_3_Al LDH thin films on stainless-steel substrate (Figure 2) was also observed. The diffraction lines of Mg_3_Al LDH thin films are sharper and more intense for the sample obtained from the silicon, confirming higher crystallinity of synthesized Mg_3_Al LDH. In both cases, the single-phase crystalline LDHs have formed [4,5,6].

The surface morphology of the representative Mg_3_Al LDH film sample obtained from the Si substrate is presented in Figure 3. The surface of the substrate is covered with a monolithic layer of agglomerated plate-like particles that are 5–10 µm in size. However, the SEM micrographs obtained at a higher magnification clearly show that these plate-like particles are composed of hexagonally shaped nanoparticles that are characteristic of LDH structures [6]. An almost identical surface morphology was observed for the LDH coatings on the stainless-steel substrate.

The sol–gel synthesis processing route for high-quality calcium hydroxyapatite coatings on silicon substrate when PVA was used as a gel-network-forming agent [67] was a source of inspiration for this study. Therefore, to enhance the sol–gel processing, the viscosity of the precursor gel was increased by adding PVAsolution, and the drying temperature was also increased.

The XRD patterns of the LDH coatings obtained from Si (Figure 4) and stainless-steel (Figure 5) substrates, however, were almost the same as without the addition of PVA.

The XRD patterns show the formation of the same crystallinity LDH phase on the Si substrate. With the dip-coating in PVA solution and drying at 300 °C (PVA’s melting point is ~266 °C), the LDH phase did not form. The LDH sample with higher crystallinity was obtained from the steel substrate. Interestingly, no side iron oxide (Fe_2_O_3_ and Fe_3_O_4_) phases were formed during the synthesis, as was observed in the case of sol–gel synthesis of calcium hydroxyapatite on stainless-steel substrate [63].

The SEM micrographs of Mg_3_Al LDH films obtained from Si and stainless-steel substrates using a precursor in the PVA solution are shown in Figure 6. The formation of nanograins of LDH is evident when PVA solution was used in the sol–gel processing. Moreover, these nanograins show a tendency to form cloudy agglomerates.

The amount of water, hydroxide, and carbonate in the formula of synthesized bulk LDH samples can be calculated from the results of thermogravimetric analyses [6,68]. For example, the composition was defined in our previous study to be [Mg_0.75_Al_0.25_(OH)_2_](CO_3_)_0.125_·4H_2_O [6]. However, the experimental procedure could be more complicated in the case of thin films and should be redefined in the future.

In Figure 7, Figure 8, Figure 9 and Figure 10, atomic force microscopy images of different Mg_3_Al LDH films prepared before and after modification on silicon and stainless-steel substrates are represented. The atomic force microscopy (AFM) data of LDH profiles were filtered with a mathematical procedure implemented in the MATLAB software. This software computes several roughness parameters at different ”walks“ of axes x (vertical) and y (horizontal) positions (see Table 1). For this reason, the AFM images were reduced and cut off from the middle 10 µm^2^ square for a better comparison. Figure 7 and Figure 8 show the Mg_3_Al LDH films dip-coated on the silicon and stainless-steel substrates, respectively.

The average RMS parameter obtained by AFM was determined to be 186.14(8) nm for the Mg_3_Al LDH surface on the silicon substrate (64.89(7) nm for raw Si substrate) and 352.62(9) nm for the Mg_3_Al LDH surface on the stainless-steel substrate (112.54(8) nm for raw Fe substrate). However, using the PVA (Figure 9 and Figure 10) solution for the modification of the Mg_3_Al LDH synthesis, the roughness increased to 733.30(8) and 1181.12(7) nm on the silicon and stainless-steel substrates, respectively. This might be because the higher concentration of polymers resulted in the formation of larger micelles of the monomer in the solution and larger polymer aggregates on the surface. As we can see from the AFM images and the calculated RMS values, the synthesized LDH coatings can be characterized as nanometer-size thin films. It was observed that the Mg_3_Al LDH film that formed on silicon substrate in the distilled water had the smoothest surface. The synthesized coatings could be applied for future work for the investigation of anticorrosive properties.

## 4. Conclusions

Mg_3_Al LDH coatings were successfully fabricated on silicon and stainless-steel substrates using the sol–gel processing route for the first time, to the best of our knowledge. The LDHs were fabricated by reconstruction of MMOs in deionized water. The MMOs were obtained by calcination of the precursor gels. The XRD patterns demonstrated the high crystallinity of the synthesized Mg_3_Al_1_ LDH coatings. The SEM micrographs clearly showed that the plate-like particles that formed on the surface are composed of hexagonally shaped nanoparticles, which are characteristic of LDH structures. The average RMS parameter obtained by AFM was determined to be 186.14(8) nm for the Mg_3_Al LDH surface on the silicon substrate and 352.62(9) nm for the Mg_3_Al LDH surface on the stainless-steel substrate. The roughness of the coatings increased to 733.30(8) and 1181.12(7) nm on the silicon and stainless-steel substrates, respectively, using the PVA solution for the modification of the Mg_3_Al LDH. The phase purity of coatings obtained from Si and stainless-steel substrates, however, was almost the same with or without the addition of PVA.

## Figures and Tables

**Figure 1 materials-12-03738-f001:**
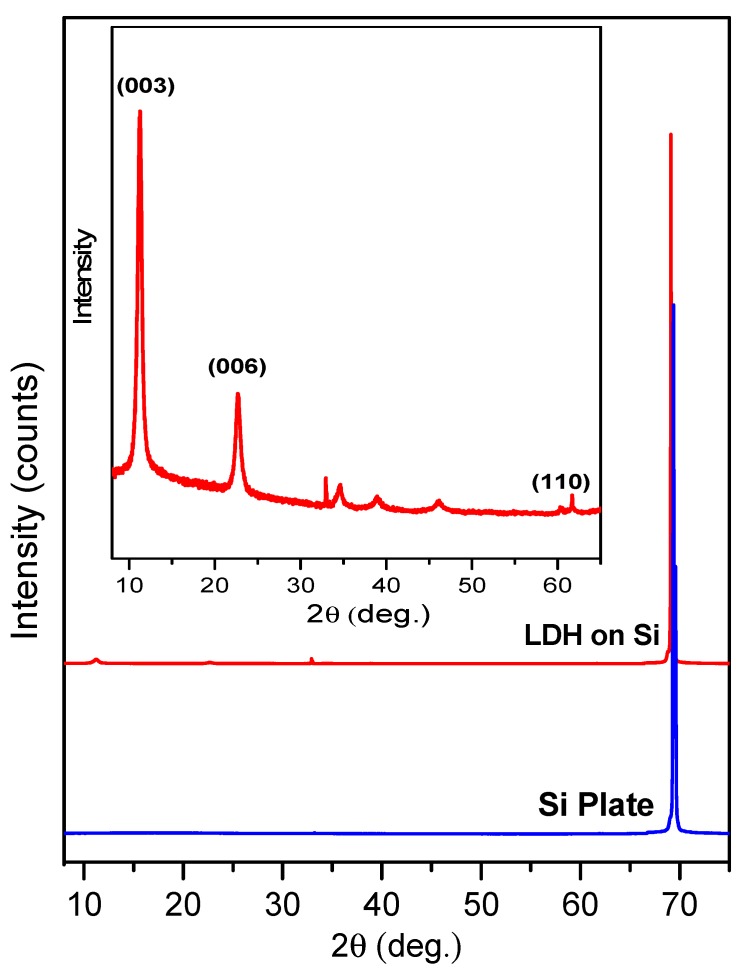
X-ray diffraction (XRD) pattern of the Mg_3_Al layered double hydroxide (LDH) coating on silicon substrate, using 15 layers of precursor at 70 °C.

**Figure 2 materials-12-03738-f002:**
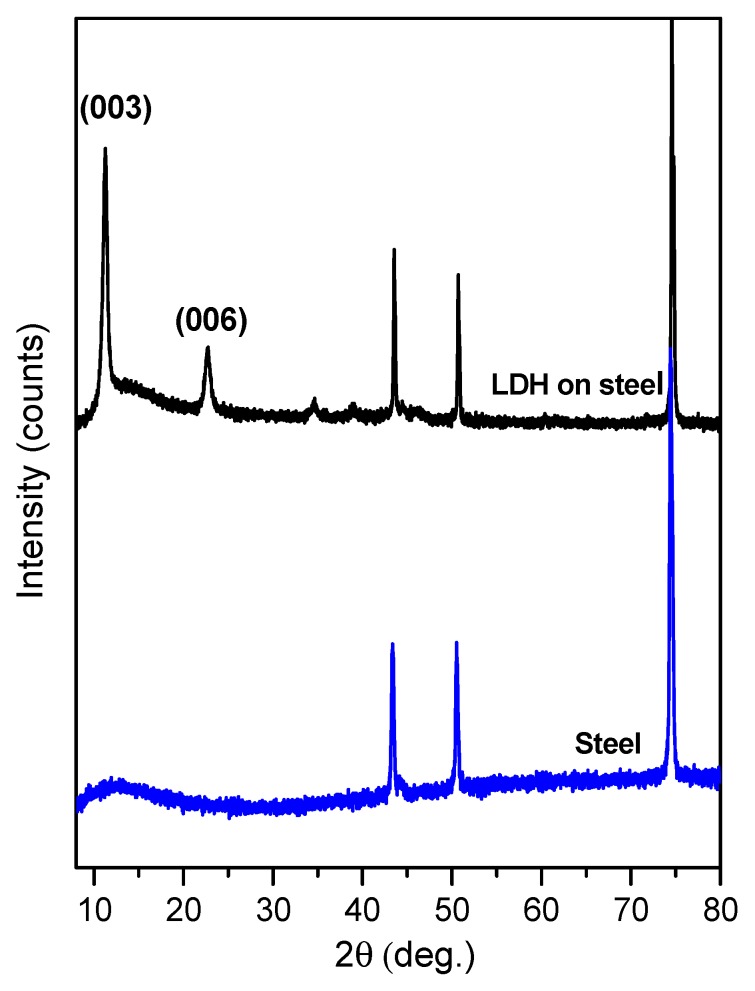
XRD pattern of the Mg_3_Al LDH coating on stainless-steel substrate, using 15 layers of precursor at 70 °C.

**Figure 3 materials-12-03738-f003:**
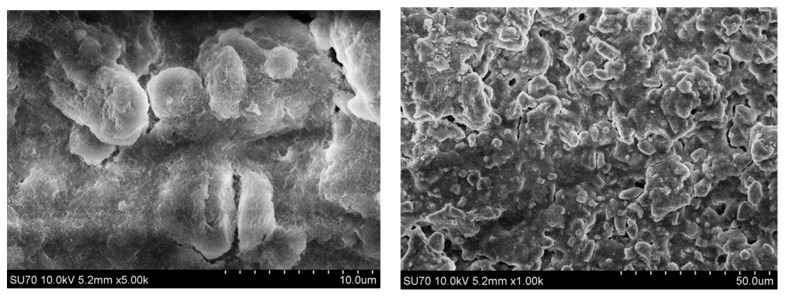
Scanning electron microscopy (SEM) micrographs of Mg_3_Al LDH film on silicon substrate, obtained at different magnifications.

**Figure 4 materials-12-03738-f004:**
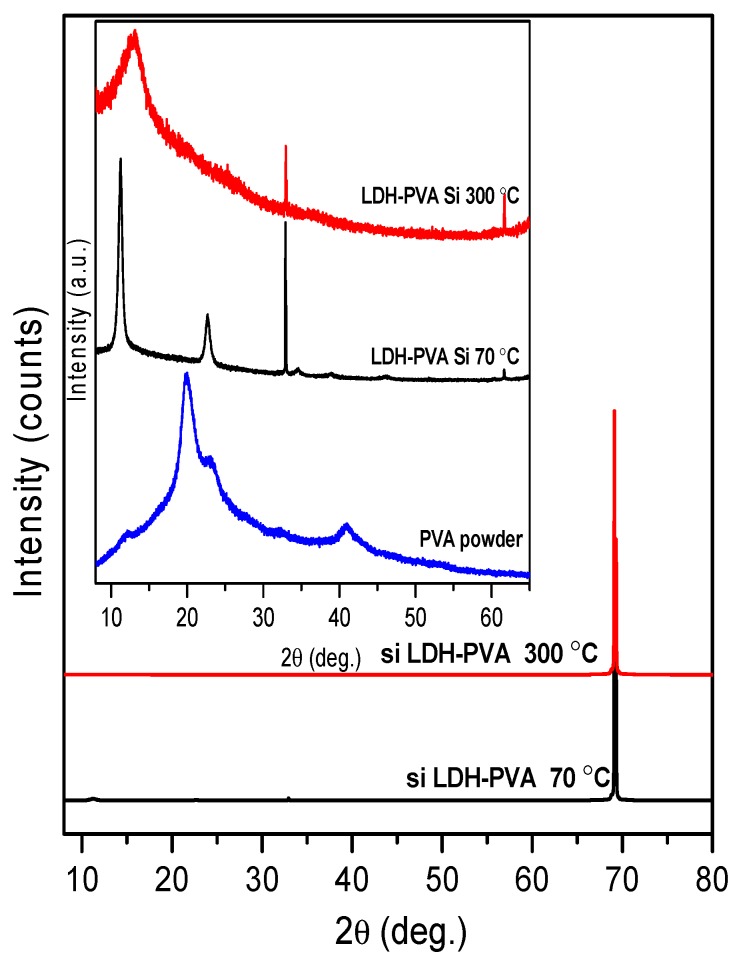
XRD patterns of the Mg_3_Al LDH coatings on silicon substrate using 15 layers of precursor with PVA solution, obtained at 70 °C and 300 °C.

**Figure 5 materials-12-03738-f005:**
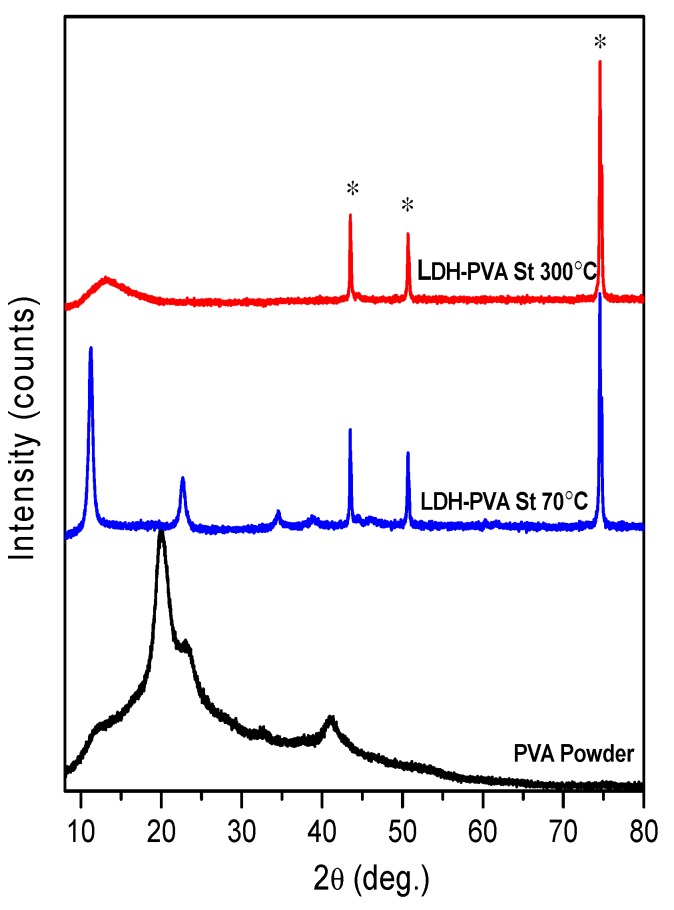
XRD patterns of the Mg_3_Al LDH coatings on stainless-steel substrate using 15 layers of precursor in PVA solution, obtained at 70 °C and 300 °C. Reflections of stainless steel are marked: *.

**Figure 6 materials-12-03738-f006:**
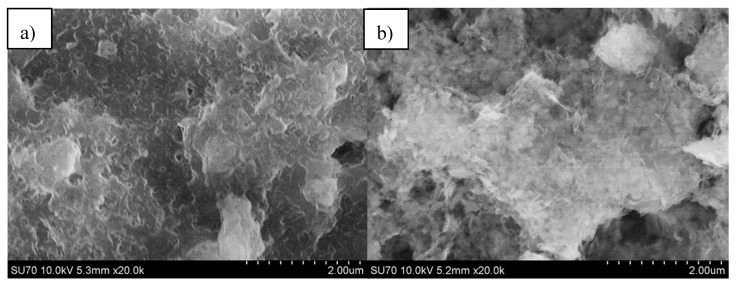
SEM micrographs of Mg_3_Al LDH films obtained on silicon (**a**) and stainless-steel (**b**) substrates in PVA solution at 70 °C.

**Figure 7 materials-12-03738-f007:**
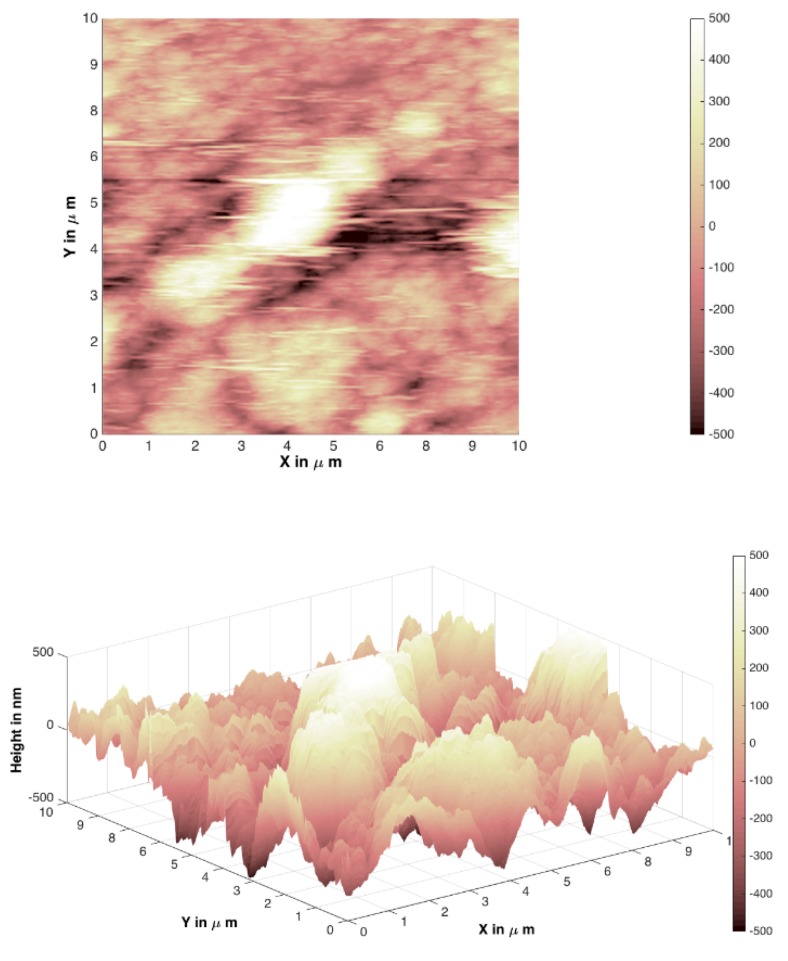
Atomic force microscopy (AFM) images of a Mg_3_Al LDH film on silicon substrate at 70 °C.

**Figure 8 materials-12-03738-f008:**
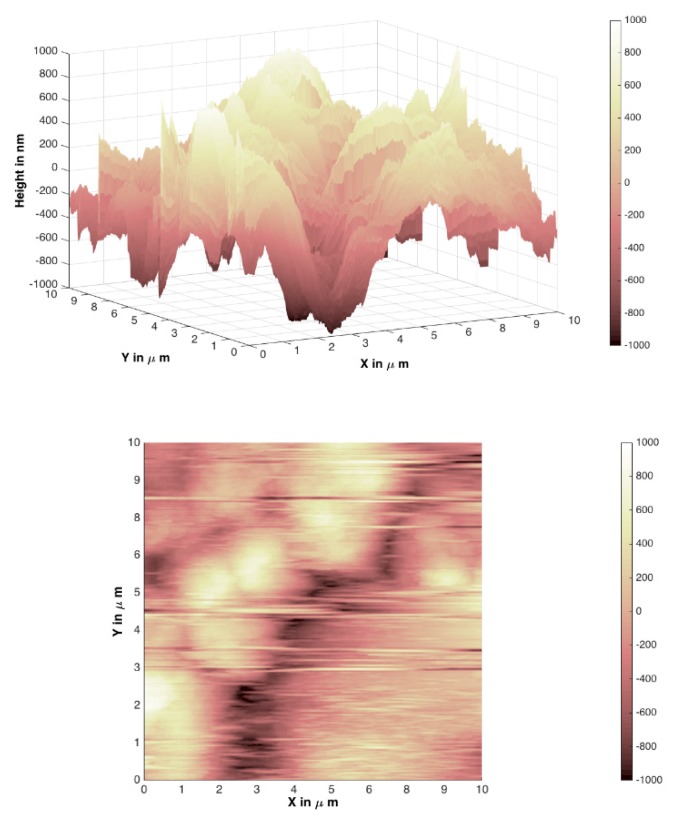
AFM images of a Mg_3_Al LDH film on stainless-steel substrate at 70 °C.

**Figure 9 materials-12-03738-f009:**
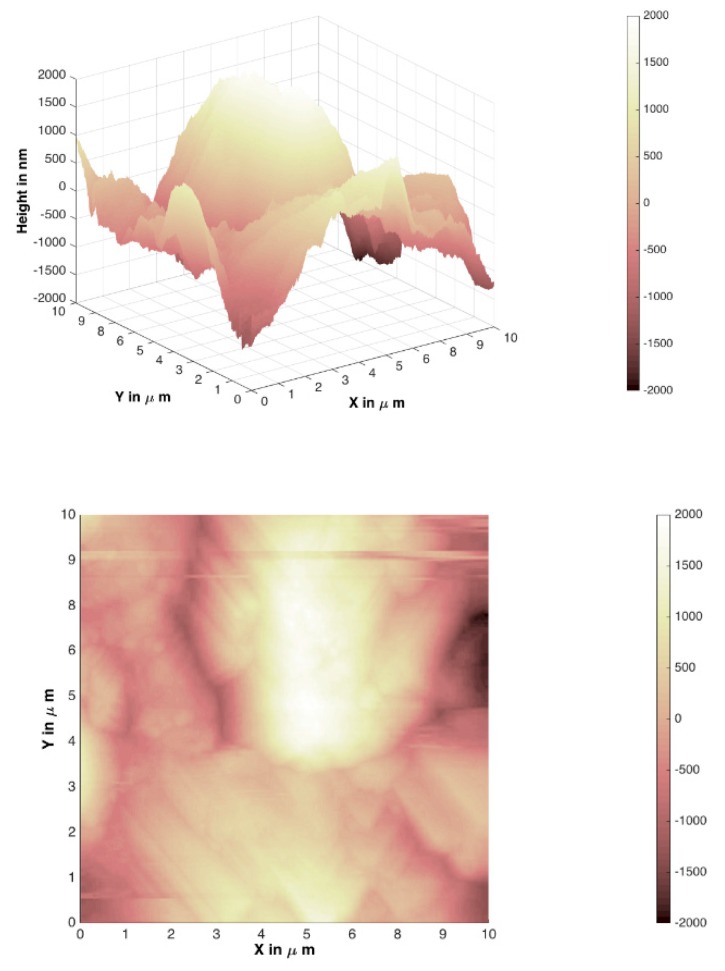
AFM images of the Mg_3_Al LDH on silicon substrate in PVA solution at 70 °C.

**Figure 10 materials-12-03738-f010:**
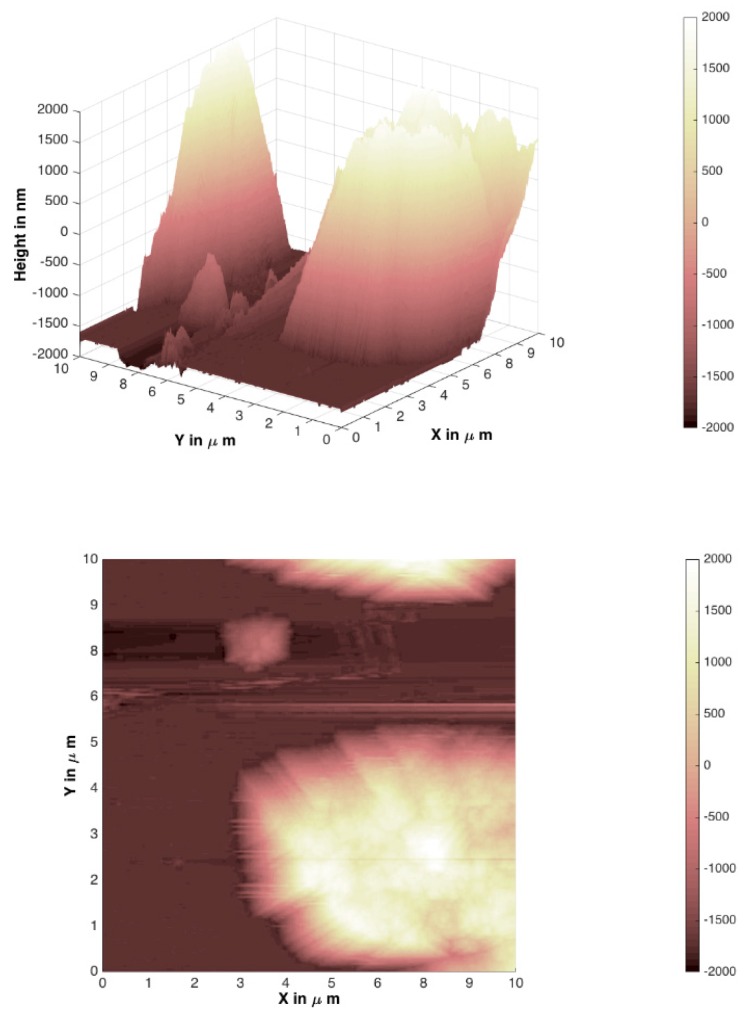
AFM images of the Mg_3_Al LDH on stainless-steel substrate in PVA solution at 70 °C.

**Table 1 materials-12-03738-t001:** The calculated root mean square (RMS) parameters from AFM images of Mg_3_Al LDH coatings, with standard deviations in parentheses.

Sample	Average RMS X (nm)	Average RMS Y (nm)	Average RMS (nm)	Min Height (nm)	Max Height (nm)	Average Heights (nm)
LDH film on silicon	172.78(8)	170.99(6)	186.14(8)	−500	500	−43.48(5)
LDH film on stainless steel	334.26(7)	345.86(7)	352.62(9)	−1000	1000	−67.55(5)
LDH film on silicon with PVA	398.66(9)	691.39(7)	733.30(8)	−2000	2000	135.03(6)
LDH film on stainless steel with PVA	774.36(8)	778.69(9)	1181.12(7)	−2000	2000	−858.89(5)

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
