# Peer review of "Sol–Gel Synthesis and Characterization of Coatings of Mg-Al Layered Double Hydroxides"

_materials, 2019, doi:10.3390/ma12223738_

Round 1

Reviewer 1 Report

In this work the authors synthetized and characterized thin films of Mg-Al layereddouble hydroxides (LDHs). The work is interesting and clear and can be considered for publication if the following revisions are considered:

Please indicate in the abstract the interest of these films. Please indicate the error bars in the values of table 1. In line 152, please correct the beginning of the sentence. In line 155, please indicate also the roughness of the silicon and of stainless-steal. Did the authors verified the effect of initial roughness of the solid support? Please comment. The figures should be placed (or diminished) in the manuscript in order to avoid blank spaces.

Author Response

The interest of these films is indicated in the Abstract: “The LDHs coatings could be used for the anticorrosion protection of different substrates, as catalysts supports and as drug delivery systems in medicine”. The standard deviations are presented in Table 1. The roughnesses of raw substrates are indicated now in the manuscript. In this study the effect of initial roughness on the solid support was not verified. The corrections are made and blank space is reduced as much as possible.

We would like to thank Reviewer for the valuable remarks. The changes made are outlined in the revised manuscript.

Reviewer 2 Report

The manuscript report on sol-gel based coating for metal surfaces. The XRD SEM and AF characterization is clear. Some notes about manuscript organization. Looking to the preparation the LDH were prepared by calcination and reconstruction. This must be stated also in the abstract and in the introduction, otherwise the reader get confused when reading the materials and methods section. Finally, it must be pointed out also in the conclusions. In the conclusions, also the role and the results obtained with PVA must be reported, with a comment if use of PVA is beneficial or not. With the previous issues solved and the following minor notes answered, the manuscript can be accepted for publication.

Minor notes:

Line 32: the LDH chemical formula must be written with apex and pedices Line 56-57: the sentence is not correct: sol-gel method are for instance cited in the review by Conterosito et al, Inorganica Chimica Acta 470 (2018) 36–50, section 2.1.2. Also the recent paper by Vieira et al Surface and Coatings Technology, Volume 375, 15 October 2019, Pages 158-163 should be cited. Line 62: There is a “\” to be deleted at the end of the sentence Line 79-80: “2°” is wrong: are the authors meaning 2theta? Then: “gradated” what’s the meaning in the sentence? All the sentence to be rewritten Line 94 “Intensive” must be “intense”. Looking to the picture LDH peaks on steel appears as sharp as those on silicon. The broadening must be commented on numbers writing down the FWHM of the peaks of LDH on both metals Line 105-106: the sentence must be changed pointing out that calcium hydroxyapatite preparation method was used as inspiration Line 121: the second chemical formula is rather strange

Author Response

The chemical formula now is written properly. The sentence and citations are correct since they reflect advantages of sol-gel processing in general not only related to LDHs. However, additional references suggested by Reviewer are also now cited. All corrections suggested by Reviewer were made and outlined.

We would like to thank Reviewer for the valuable remarks. The changes made are outlined in the revised manuscript.

Reviewer 3 Report

The manuscript „Sol-gel synthesis and characterization of coatings of Mg-Al layered double hydroxides (LDHs)”by A. Smalenskaite et al. studies new synthetic approaches for preparation of thin films of Mg-Al layered double hydroxides. The films are characterized by XRD, SEM and AFM.
This topic is interesting for publication in Materials. The manuscript needs minor revisions before recommending it for publication.
1. Line 32: Chemical formula not correctly formatted with subscripts.
2. Line 62: Remove \
3. Lines 65 and 68: ml instead of mL
4. Figure 3: The pictures should be arranged side by side as in Fig. 6
5. Figures 7, 8, 9, 10: The AFM images should be arranged side by side. One of the color scales can be removed since they are the same. When there are two pictures in one figure, they should be labeled a) and b).

Author Response

The chemical formula now is written properly. Correction is done. Corrections are done. Fig. 3 is modified according to the suggestion of Reviewer. Such arrangement of AFM images was done by Editorial members. The original version was as Reviewer mentioned.

We would like to thank Reviewer for the valuable remarks. The changes made are outlined in the revised manuscript.

Reviewer 4 Report

Dear authors 

the work is interesting and the paper is in my opinion of good quality, I would recommend to check the English and text as there are some mispelling (it is "average" not "avarage") and some missing letter (see line 152). I would suggests to add a low magnification image of the un-coated and coated sample, as to show the homogeneity of it.

Author Response

The grammar is checked once again. In our opinion the SEM micrographs of substrate would be redundant information. In many publications these images are already presented. For instance (sorry, I cannot insert image).

We would like to thank Reviewer for the valuable remarks. The changes made are outlined in the revised manuscript.